# Distorted TCR repertoires define multisystem inflammatory syndrome in children

**Amna Malik**[1☯], **Eszter N. Tóth**[2☯], **Michelle S. Teng**[2], **Jacob Hurst**[2], **Eleanor Watt**[3], **Lauren Wise**[4], **Natalie Kent**[4], **Jack Bartram**[5], **Louis Grandjean**[6], **Margarita Dominguez-Villar**[7], **Stuart Adams**[4‡], **Nichola Cooper**[1‡*]

**1** Department of Immunology and Inflammation, Centre for Haematology, Imperial College London, London, United Kingdom, **2** Etcembly Ltd, Magdalen Centre, Robert Robinson Way, Oxford, United Kingdom, **3** Molecular and Cellular Immunology Department, UCL Great Ormond Street Institute of Child Health, London, United Kingdom, **4** SIHMDS-Haematology, Great Ormond Street Hospital for Children, London, United Kingdom, **5** Department of Haematology, Great Ormond Street Hospital for Children, London, United Kingdom, **6** Paediatric Infectious Diseases, Great Ormond Street Hospital for Children, London, United Kingdom, **7** Department of Infectious Diseases, Imperial College London, London, United Kingdom

☯ These authors contributed equally to this work.
‡ SA and NC also contributed equally to this work.
* n.cooper@imperial.ac.uk

**Data Availability Statement:** The raw FastQ files are deposited at the Short Read Archive (https://www.ncbi.nlm.nih.gov/sra) under accession number SRP136075.

## Abstract

While the majority of children infected with severe acute respiratory syndrome coronavirus 2 (SARS-CoV-2) display mild or no symptoms, rare individuals develop severe disease presenting with multisystem inflammatory syndrome (MIS-C). The reason for variable clinical manifestations is not understood. Here, we carried out TCR sequencing and conducted comparative analyses of TCR repertoires between children with MIS-C (n = 12) and mild (n = 8) COVID-19. We compared these repertoires with unexposed individuals (samples collected pre-COVID-19 pandemic: n = 8) and with the Adaptive Biotechnologies MIRA dataset, which includes over 135,000 high-confidence SARS-CoV-2-specific TCRs. We show that the repertoires of children with MIS-C are characterised by the expansion of TRBV11-2 chains with high junctional and CDR3 diversity. Moreover, the CDR3 sequences of TRBV11-2 clones shift away from SARS-CoV-2 specific T cell clones, resulting in distorted TCR repertoires. In conclusion, our study reports that CDR3-independent expansion of TRBV11-2+ cells, lacking SARS-CoV-2 specificity, defines MIS-C in children.

## Introduction

Coronavirus disease 2019, COVID-19, caused by the novel severe acute respiratory syndrome coronavirus 2 (SARS-CoV-2) is associated with high morbidity and mortality in older individuals, and in those with additional comorbidities. In contrast, children represent a small proportion of COVID-19, comprising less than 2% of cases worldwide [1, 2]. While most children with COVID-19 are asymptomatic or present with mild disease, rare individuals develop severe disease presenting with multisystem hyperinflammatory syndrome (MIS-C) including persistent fever, severe abdominal pain, diarrhoea, myocardial dysfunction, cardiogenic shock,

**Funding:** N.C and M.D.V. are partially funded by Imperial College NIHR BRC. A.M. is funded by the Jon Moulton Charity. The funders had no role in study design, data collection and analysis, decision to publish, or preparation of the manuscript.

**Competing interests:** E.N.T., M.S.T. and J.H. have a financial interest in Etcembly Ltd. The other authors declare no competing interest. This does not alter our adherence to PLOS ONE policies on sharing data and materials.

rash and neurological disorders [3, 4]. This disparity in symptoms is not understood but could represent a difference in the T cell response to SARS-CoV-2.

Whilst it is clear that the adaptive immune response plays an important part in clearance of SARS-CoV-2 infection [5–7], the exact role of T cells in the resolution or potential exacerbation of SARS-CoV-2 infection is not known [8]. A large number of unexposed individuals have SARS-CoV-2 reactive CD4+ memory T cells and these memory T cells have been shown to exhibit cross-reactivity against seasonal "common cold" coronavirus strains [8–11]. In addition, studies to date have shown that T cell responses develop in almost all patients with confirmed SARS-CoV-2 infection [12] and remain detectable for several months following infection [8]. In adult COVID-19 patients CD8$^+$ T Cell activation status evolves with disease severity in a non-monotonous way [13]: effector-like cell clusters expand in mild disease and fall during severe disease with the highest level of T-cell polyfunctionality in moderately ill patients. In children activation and proliferation level of CX3CR1+ CD8+ T-cells is much higher in multisystem inflammatory syndrome compared to mild COVID-19 [14]. These cells can interact with fraktalkine-expressing activated endothelium and patrol vasculature, thus this interaction might explain the cardiovascular involvement in children with multisystem inflammatory syndrome. Polyclonal expansion of TCR Vb 21.3+ (TRBV11-2) CD4+ and CD8 + T cells have been shown to be associated with MIS-C [15–17].

T cell receptor sequencing allows the detection and quantification of specific T cell clones and enables us to capture unique patient TCR repertoires. We hypothesized that comparing the TCR repertoire in children with MIS-C or mild COVID-19 and contrasting with either unexposed individuals or COVID specific data sets, could reveal TCR repertoire features that could help understand features associated with MIS-C in children.

## Materials and methods

This study was done in accordance with The Multi Centre Research Ethics Committee in Wales guidelines MREC Wales reference 06/Q0508/16. Written consent was obtained for all participants.

### TCR sequencing

Next generation sequencing of the T-cell receptor (TCR) was carried out as previously described [18]. Briefly, DNA was extracted from patient blood samples using DNeasy Blood & Tissue kit (Qiagen), quantified using a Qubit Fluorometer (ThermoFisher Scientific) and amplified by multiplex-PCR of rearranged variable, diverse, joining (VDJ) segments of the TCR genes, which encode the hypervariable CDR3 domain. The products were size selected using Pronex beads (Promega) and subsequently sequenced on a MiSeq (Illumina).

Analysis of the raw TCR sequences was performed using MiXCR [19]. A built-in library of reference germline V, D, J, and C gene loci from the ImMunoGeneTics (IMGT) database (imgt.org) is employed by MiXCR. The IMGT nomenclature for TCR gene segments is used throughout the study.

**MIRA data set.**  Adaptive Biotechnologies have created a MIRA dataset of T cell clones, which includes over 135,000 high-confidence SARS-CoV-2-specific TCRs (MIRA clones) at the time of this study [20, 21]. The MIRA assay is a high-throughput multiplex tool that maps TCRs binding to SARS-Cov2 virus epitopes by exposing PBMCs to Sars-Cov2 minigenes or peptide pools, sorting T-cells based on surface expression of activation markers and sequencing the TCRs expressed by these activated T-cells. The dataset is a collection of matching antigen-TCR data from both Sars-Cov2-convalescent subjects and unexposed individuals. The

antigens include Sars-Cov2 epitopes presented by a diverse set of MHC class I as well as MHC class II alleles, thus capturing response by CD8+ and CD4+ T-cells

## Data analysis and visualization

All visualization and standard statistical analysis were conducted using R version 4.0.3 and Python 3.9.1. Plots were generated using the ggplot2 R package [22]. Correlation was quantified by Spearman's rank correlation coefficient ($\rho$). Paired analyses were performed by nonparametric paired Wilcoxon test. All tests were performed two-sided with a nominal significance threshold of P < 0.05. In all cases of multiple comparisons, false discovery rate (FDR) correction was performed using the Benjamini-Hochberg procedure. Illustrations were prepared with the BioRender package.

## TCR repertoire analysis

TRBV gene usage, TRBV-TRBJ junction frequencies and repertoire global metrics were calculated for each sample. When samples from multiple time points were available for a patient, the mean values were used for downstream analysis. Differential gene expression analysis of TRBV genes was conducted using the EdgeR package [23].

The normalized repertoire richness (R) was calculated as follows:

$$R = \frac{n}{r} * 10^6$$

Where n is the number of clonotypes, and r is the number of reads in a repertoire.

The Shannon evenness index (J') was calculated as in (Attaf et al. 2018) [24]: $J' = \frac{H'}{\ln(n)}$

$$H' = -\sum_{i=1}^{n} p_i \ln(p_i)$$

Where $p_i$ is the frequency of the $i^{th}$ clonotype in a population of $n$ clonotypes. $J'$ is undefined for monoclonal samples. Low $J'$ values approaching 0 indicate minimal evenness such as after clonal expansion of antigen-specific clones. The maximum value of $J$ is 1, when all clonotypes have equal frequencies, thus the population is perfectly even.

## Network analysis

First, multiple samples from the same patient collected at different time points were combined. Next, to adjust for different sequencing coverage and clonal depth among patients we randomly down sampled each repertoire to 1,000 clones. Next, we calculated pairwise amino acid sequence similarity of these 1,000 clones by constructing Levenshtein distance (LD) matrices of their CDR3 sequences using the stringdist package [25]. Networks were generated using the igraph package [26]. Each node in the TCR similarity network represents a unique amino acid clone, and edges between nodes are constructed by connecting nodes that differ by no more than 2 amino acids (LD1-2) in their CDR3 sequences. The final similarity network contains only nodes that make at least one connection to another node in the network. Clusters were defined as groups of interconnected nodes. Classic graph analysis metrics [27], such as the number of nodes, edges, node degree and cluster sizes were calculated using the igraph package as well. The random repertoire down sampling, network construction and calculation of network metrics were repeated 10 times for each patient and the mean values were used for the downstream analysis.

### Distance to MIRA analysis

First, samples from the same patients taken at different time points were combined, and MIRA clones in the repertoires were found by mapping the CDR3 sequences to the Adaptive MIRA database [20, 21]. Next, Levenshtein distance matrices (LD) were constructed between all clones and all MIRA clones in a repertoire using the stringdist package [25]. The *distance to MIRA* value was defined as the shortest distance a MIRA clone. Gene-wise *distance to MIRA* distributions were analysed for those TRBV genes that occupied at least 1% of mean expressed repertoire in each patient cohort. These distributions were fitted with Gaussian probability density functions and the difference between patient cohorts was assessed by comparing a model fitting all symptom groups together to a model taking into account the differences between symptom groups. The difference between the two models was determined with ANOVA analysis (as described in [28]), with all p-values corrected for multiple hypothesis testing using Benjamini-Hochberg adjustment.

## Results and discussion

### Cohort characteristics (Table 1)

The cohort included children at Great Ormond Street Hospital, London, UK with PCR confirmed SARS-CoV-2 that had:

1. asymptomatic infection, cough, or fever, defined as mild disease patients(n = 8),

2. multisystem hyperinflammatory syndrome (MIS-C) (n = 12)

Of the patients tested (n = 14), 86% were SARS-CoV-2 antibody positive. Half of the patients (50%, n = 10) had pre-existing co-morbidities: 86% (n = 7) of the mild disease patients

**Table 1. Cohort characteristics of SARS-CoV-2 infected children and unexposed children.**

| Column1 | Unexposed (n = 8) | Mild (n = 8) | MIS-C (n = 12) |
|---|---|---|---|
| **Age in months (IQR)** | 84(120) | 96 (87) | 60 (51) |
| **SARS-CoV-2 PCR Positivity** | N/A | 100%(2/2) | 100%(12/12) |
| **Antibody Test positivity** | N/A | 86%(12/14) | 83%(10/12) |
| **Longitudinal samples** | N/A | 63%(5/8) | 8%(1/12) |
| **Symptoms** | | | |
| Fever | N/A | 5%(1/12) | 0 |
| Cough | N/A | 5%(1/12) | 0 |
| CNS Disease | N/A | 0 | 42%(5/12) |
| Hyperinflammatory Syndrome | N/A | 0 | 58%(7/12) |
| None | N/A | 75%(6/8) | 0 |
| **Other co-morbidities** | | | |
| Dravet Syndrome | NA | 5%(1/8) | 0 |
| hepatoblastoma | NA | 0 | 5%(1/12) |
| Neuroblastoma | NA | 5%(1/8) | 0 |
| TAPDV | NA | 0 | 5%(1/12) |
| Tonsillitis | NA | 0 | 5%(1/12) |
| Atypical teratoid rhabdoid tumour | NA | 5%(1/8) | 0 |
| PNET metastatic tumour | NA | 5%(1/8) | 0 |
| B cell ALL | NA | 10%(2/8) | 0 |
| Complex metabolic disorder | NA | 5%(1/8) | 0 |
| None | 100%(10/10) | 13%(1/8) | 70%(9/12) |

and 25% (n = 3) of MIS-C patients. The majority of the mild disease patients (n = 6, 75%) and 25% (n = 3) of MIS-C patients were lymphopenic at the point of sampling. The detailed patient information is shown in Table 1.

A control cohort included DNA samples taken from children who had no exposure to SARS-CoV-2 and were collected before the COVID-19 pandemic (n = 8).

TCR sequencing and repertoire analysis was performed using bulk DNA extracted from blood samples (Fig 1a). TCR repertoire metrics can be found in S1 Table.

## The repertoires of children with MIS-C are characterized by the expansion of TRBV11-2

We performed principal component analysis (PCA) of TRBV gene usage to determine their global distributions between patient groups with different symptom severity. Patient cohorts showed separation along the first principal component in children with MIS-C clustering clearly apart from the children with mild disease (Fig 1b).

To better characterise TRBV gene expression skewing in children with MIS-C, we performed differential gene expression analysis of TRBV genes between patient cohorts. This revealed that the expansion of TRBV11-2 chain is mostly responsible for the different TRBV gene usage pattern of children with MIS-C (Fig 1c). This is in line with previous studies that report TRBV11-2 expansion of CD4+ and CD8+ T cells as a hallmark of MIS-C [15–17]. The fraction of TRBV11-2 in a repertoire did not show correlation with the patient's age, the number of days since COVID-19 diagnosis or the antibody status (S1 Fig).

## TRBV11-2 has high junctional diversity in all patient cohorts

To interrogate whether the expansion of TRBV11-2 in children with MIS-C is associated with a specific CDR3b motif of J genes usage, we analysed the junctional diversity of TRBV11-2 chains. We compared the frequencies of rearranged J genes in children with mild symptoms to children with MIS-C. We did not observe a difference between the cohorts indicating that the expansion of TRBV11-2 in children with MIS-C was not driven by clones harbouring specific TRBV-TRBJ junctions, and that TRBV11-2 had high junctional diversity in all our patient cohorts (Fig 1d). The polyclonal nature of the TRBV11-2 expansions has been shown before in MIS-C [15]. The alignment of expanded TRBV11-2 CDR3 sequences from MIS-C patients did not reveal the presence of an enriched CDR3 sequence motif, thus we concluded that the expansion of this V-gene was unrelated to the sequence of the CDR3 peptide binding motif (Fig 1e). This finding is in line with two recent studies suggesting that a superantigen-like sequence motif highly similar to staphylococcal enterotoxin B (SEB) near the S1/S2 cleavage site of the SARS-CoV-2 spike (S) protein can interact with the CDR2 region of TRBV11-2 and may be able to form a ternary complex with MHCII [29, 30]. A superantigen-like interaction bypasses the antigen-specific CDR3 region and involves only the constant CDR2 region, thus all T-cells expressing a given TRBV gene expand regardless of their peptide-MHC specificity. This can lead to the domination of the TCR repertoire by the superantigen-interacting V-gene, while the virus-specific T-cells fail to expand and respond to the infection [31]. In addition, some of the TCRs harbouring the superantigen-interacting V-gene can be autoreactive causing severe autoimmune reaction in the patient. To this date, the presence of autoantibodies in adults suffering from MIS-C has been reported [32, 33], but little is known about the involvement of autoreactive T-cells in the development of severe disease in adults or MIS-C in children. It is possible to hypothesize that if TRBV11-2 TCRs interact with the SARS-CoV-2 spike (S) protein consistent with a superantigen event, this can trigger the expansion and activation of autoreactive TCR clones causing life-threatening complications.

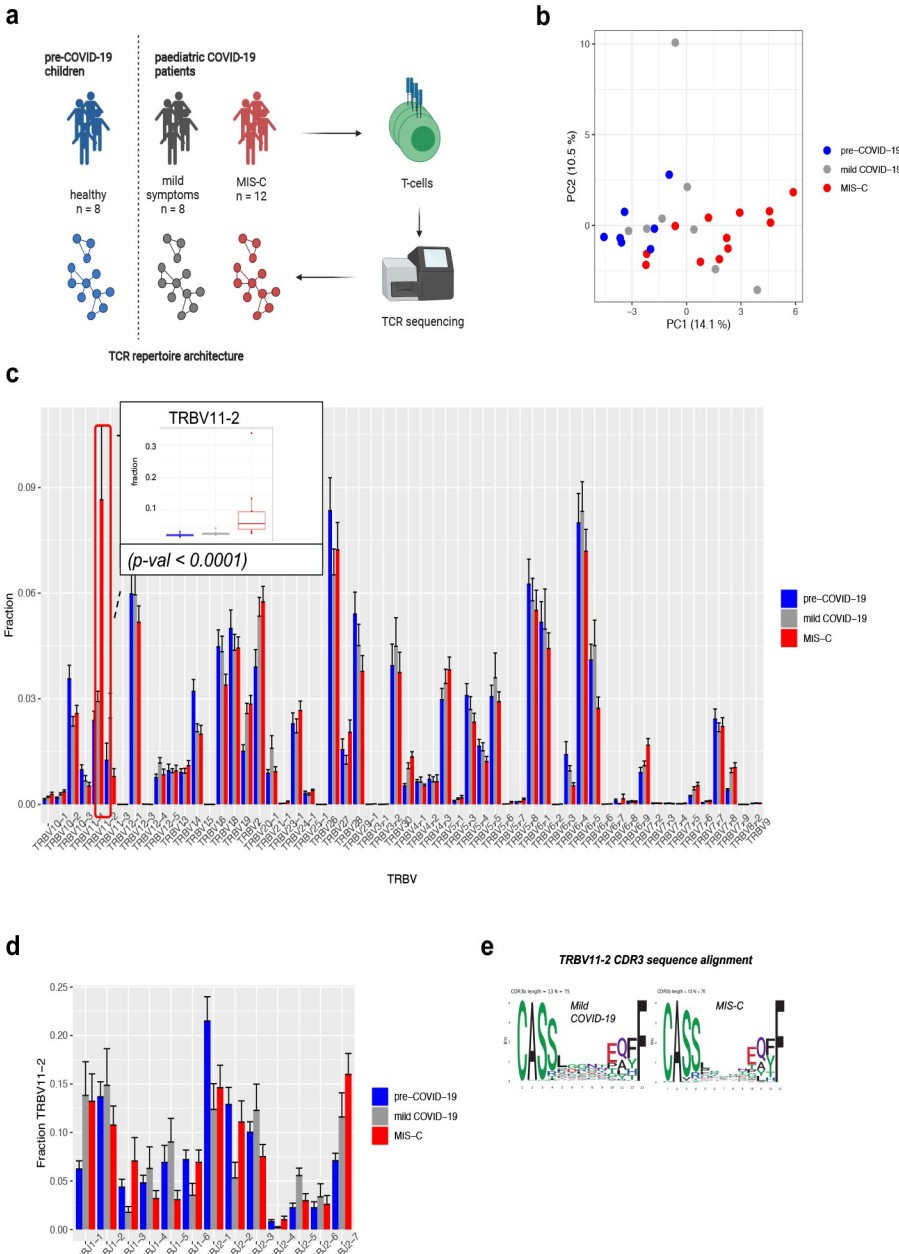

**Fig 1. TRBV11-2 is expanded in the repertoire of children with MIS-C.** (a) Overview of study design. (b) Principal component analysis (PCA) of differential TRBV usage in children with MIS-C (n = 12), children with mild symptoms (n = 9), and pre-COVID-19 children (n = 8). Values in brackets show the percentage of variation explained by each principal component. (c) Frequencies of TRBV genes in the patient cohorts. Bars indicate mean + SEM. Insert: TRBV11-2 usage in patient cohorts. Differential gene expression analysis of TRBV genes was conducted using the EdgeR package. TRBV11-2 is expanded in MIS-C compared to children with mild symptom (p-val < 0.0001). (d) Differential usage of J genes rearranged with TRBV11-2. Bars indicate mean + SEM. TRBV11-2 rearrangement frequencies were analysed using the EdgeR package. (e) CDR3 diversity of TRBV11-2 in children with mild disease and MIS-C displayed as positional weight matrix. CDR3 sequences containing 13 amino acids are shown as examples.

## The expansion of TRBV11-2 in children with MIS-C does not alter the TCR repertoire's overall architecture

We assessed the effect of MIS-C on the children's TCR repertoires by computing global T-cell metrics. We found that children infected with SARS-CoV-2 had a lower number of TCR clones than healthy ones, but there was no correlation between the number of clones and symptom severity within the disease cohort indicating that the expansion of TRBV11-2 did not influence the repertoire's richness (Fig 2a). The children's age, number of days after COVID-19 diagnosis or presence of co-morbidities did not affect the repertoire's richness either (S2 Fig). All repertoires were diverse as shown by the high values of Shannon evenness index (Fig 2b). This finding might seem unexpected considered that the TRBV11-2 gene was significantly expanded in children with MIS-C, however, as shown above (Fig 1d and 1e), the expanded TRBV11-2 chains had high CDR3 sequence diversity, thus they contributed to the repertoire's overall diversity and richness. This supports the hypothesis that TRBV11-2 chains might engage in a superantigen-binding interaction with the Sars-Cov2 S-protein regardless of their CDR3 sequence [29, 30].

To gain insight into the architecture of patient repertoires we applied a network analysis approach (Fig 2c). This method has been used before to decipher the overall connectivity structure of antibody and TCR repertoires [34–36]. Upon antigen exposure, specific T-cells expand, thus they are more likely to be captured when blood samples are collected, and they are easier to detect by TCR sequencing. Homologous TCRs with highly similar CDR3 sequences often recognise the same antigens [37, 38], resulting in highly connected TCR sequence similarity networks, with the most similar sequences forming clusters. We analysed connectivity levels of our patient TCR networks by comparing their graph metrics. Usually, high number of nodes and edges in a network, large clusters and high number of connections per node indicate highly connected networks with the presence of many similar TCR sequences. We found that patients with MIS-C had slightly higher level of network connectivity indicated by the higher number of nodes. The number of network edges, the mean degree or the size of the largest cluster did not change significantly, thus the level of increase in network connectivity was small. This increase might be explained by the presence of more similar clones that arise in response to MIS-C and prolonged antigen exposure. Overall, the patient repertoires were robust, SARS-CoV-2 infection and the expansion of TRBV11-2 chains in the MIS-C cohort did not cause a major change in the immune networks' overall structure (Fig 2d).

## TRBV11-2 clone sequences of children with MIS-C shift away from the COVID-19-specific clones

By mapping the clones to the publicly available Adaptive MIRA dataset (described in methods) we can identify TCRs with potential to be SARS-Cov2-specific [20, 21]. This database contains more than 130,000 unique TCRβ sequences with known specificities to SARS-CoV-2 antigens from a high number of donors with diverse HLA-backgrounds (MIRA clones). We found that both class I and class II MIRA clones were ubiquitously expressed in our patient repertoires, and the number of identified MIRA clones in a repertoire was directly proportional to the total number of clones (S3 Fig). To assess the overall level of similarity to MIRA clones in a repertoire we defined the *distance to MIRA* measure for each clone (Fig 3a). Briefly, a distance matrix was constructed between all clones and all MIRA clones in a repertoire, and the distance to the closest MIRA clone was identified. Naturally, MIRA clones will have a *distance to MIRA* score of 0.

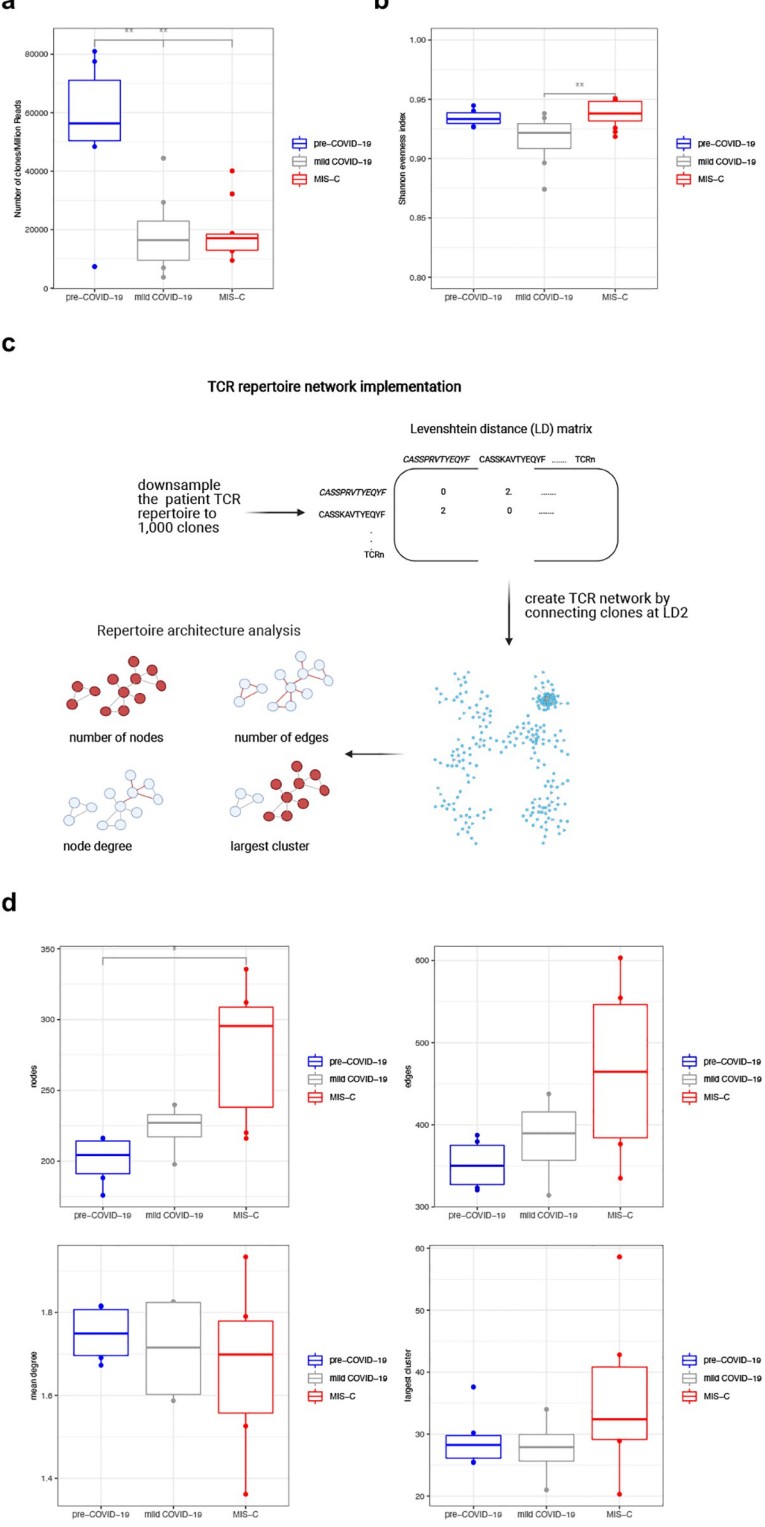

**Fig 2. The expansion of TRBV11-2 in children with MIS-C doesn't alter the TCR repertoire's overall architecture.**
(a, b) TCR repertoire metrics of patients. Significance determined by unpaired Wilcoxon test between each pediatric group, with adjustment for multiple comparisons using Benjamini-Hochberg correction, indicated by: * p<0.05, ** p<0.01, and *** p<0.001. Lack of notation for specified comparisons indicates no statistical significance. (c) Overview of sequence similarity network analysis: Each repertoire was downsampled to 1,000 clones. Pairwise amino acid

sequence similarity of these 1,000 clones was calculated by constructing Levenshtein distance (LD) matrices of their CDR3 sequences. Each node in the TCR similarity network represents a unique amino acid clone, and edges between nodes are constructed by connecting nodes that differ by no more than 2 amino acids (LD1-2) in their CDR3 sequences. Classic graph metrics, such as node and edge number, node degree and the size of the largest cluster were calculated. The random down sampling and network construction were repeated 10 times and the mean value of the network metrics was used for the downstream analysis. (d) TCR sequence similarity network metrics of patient cohorts. Significance determined by unpaired Wilcoxon test between each paediatric group, with adjustment for multiple comparisons using Benjamini-Hochberg correction, indicated by: * p<0.05, ** p<0.01, and *** p<0.001. Lack of notation for specified comparisons indicates no statistical significance.

Distance to class I and class II MIRA distributions were fitted with Gaussian probability density functions and the difference between patient cohorts was assessed by comparing a model fitting all symptom groups together to a model considering the differences between symptom groups. The difference between the two models was determined with ANOVA analysis, with all p-values corrected for multiple hypothesis testing using Benjamini-Hochberg adjustment. Solid curves show the probability distribution functions determined by fitting the patient cohorts separately, shaded areas show 75% confidence intervals. P-values denote the statistical assessment of the models fitting all patient cohorts together versus fitting each patient cohort separately. See the distance to MIRA distributions of clones with TRBV gene other than TRBV11-2 in S4 and S5 Figs.

To interrogate if the properties of TRBV11-2 clones change in the repertoires of children with MIS-C compared to children with mild symptoms, we plotted their *distance to MIRA* distributions. In children with mild symptoms, most TRBV11-2 clones have a CDR3 sequence 3–4 amino acids different from a class I MIRA clone (Fig 3b). The distribution shifts to higher *distance to MIRA* values in children with MIS-C, the maximum density being around *distance to MIRA* 4–5 accompanied by a longer tail in the region on high *distance to MIRA* values. We fitted the curves with Gaussian probability distribution functions and achieved significantly better fit when symptom severity was considered, indicating that there is a significant

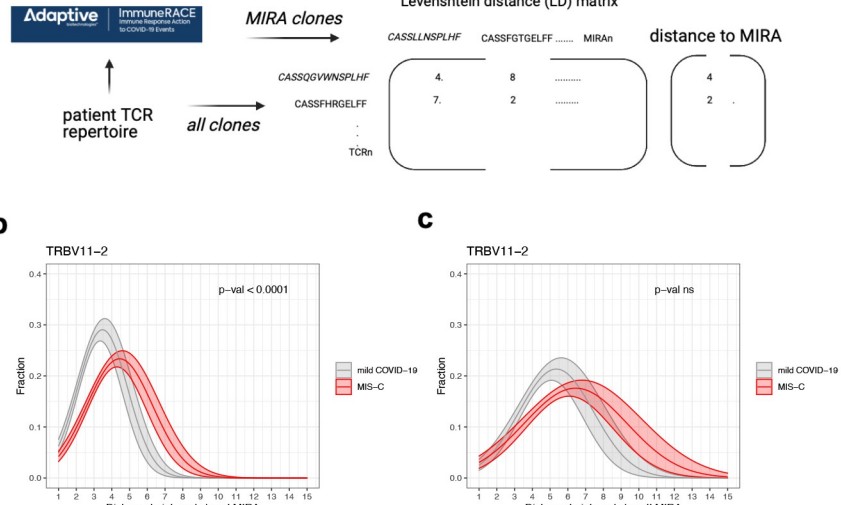

**Fig 3. Distance to MIRA analysis.** (a) Overview of distance to MIRA analysis: A distance matrix was constructed between all clones in a repertoire and those that mapped to the Adaptive MIRA database, and the distance to the closest MIRA clone was identified. (b) Distance to class I MIRA distribution of TRBV11-2 clones in the repertoires of children with mild COVID-19 or MIS-C. (c) Distance to class II MIRA distribution of TRBV11-2 clones in the repertoires of children with mild COVID-19 or MIS-C.

difference in the distance to MIRA distributions of TRBV11-2 clones in children with mild disease and MIS-C (Fig 3b). All other TRBV chains had the same distance to class I MIRA distribution in mild disease and MIS-C (S4 Fig).

TCRs recognizing the same antigen often have highly similar CDR3 sequences [21, 38], therefore, TRBV11-2 clones shifting away from the Sars-Cov2-specific MIRA hits in children with MIS-C suggests that these clones might be less effective at binding SARS-Cov-2 antigens. In children with MIS-C, TRBV11-2 chain is highly expanded with no CDR3 motif expansion or specific J-gene usage and the CDR3 sequences of these expanded TRBV11-2 clones shift away from the SARS-CoV-2-specific MIRA hits. This suggests that the expansion of TRBV11-2 clones might be independent of the classical CDR3-peptide-MHC-mediated antigen recognition. As SARS-CoV-2 Spike protein has been suggested to have a superantigen structure [29, 30] and considering our findings, it is likely that the observed TCR repertoire skewing is superantigen-induced which results in the repertoire being dominated by the expression of non-specific TCRs that we speculate are unable to respond to the infection and instead contributing to the hyperinflammatory state.

There are some striking clinical similarities between MIS-C and toxic shock syndrome (TSS), caused by bacterial superantigens [39, 40]. Superantigens simultaneously bind major histocompatibility complex (MHC) class II (MHCII) molecules on antigen presenting cells and T cell receptors (TCRs) of both CD4+ and CD8+ T cells [41]. They can circumvent TCR specificity by binding to specific TCR β-chains in a complementary-determining region 3 (CDR3)-independent manner, resulting in broad T cell activation. In patients with MIS-C, skewing of specific TCR β Variable (V) genes, with diverse CDR3 and Joining (J) usage, has been reported to correlate with disease severity, consistent with superantigen triggered immune activation [30, 42].

The distance distribution of TRBV11-2 clones to class II MIRA hits showed a slight, but not significant shift to higher values (Fig 3c). Although the shift in the median values is evident, the high variation in the data didn't allow us to accurately estimate the distribution curves, thus our model did not determine significant difference between mild and MIS-C patients. The distance to class II MIRA distributions of all other TRBV genes did not show any significant difference between our patient cohorts either (S5 Fig). The high variation in the data can be explained by the low number of class II MIRA hits in our repertoires. Class II MIRA clones accounted for about 0.4% of all clones in a repertoire (S3 Fig), whereas about 5% of each repertoire was a class I MIRA clone (S3 Fig). This discrepancy can be explained by the fact that most of the Adaptive MIRA database consists of class I hits. The MIRA database contains several data releases with most of them reporting TCRs that interact with antigens in the context of MHC class I.

In summary, the T cell repertoire of children with MIS-C is distorted, by TRBV11-2+ T cell clonal expansion and activation, which could be super antigen induced causing an aberrant immune response and leading to clinical manifestations reminiscent of toxic shock syndrome [39, 40]. We report the use of two metrics to define the severity of disease in children infected with SARS-CoV-2: 1) CDR3-independent expansion of TRBV11-2+ T cells, 2) a lack of SARS-CoV-2 specificity in TRBV11-2+ T cells, measured by distance to Sars-Cov2-specific MIRA clones. These two metrics can serve as biomarkers for early detection of multisystem inflammatory syndrome in children (MIS-C) guiding physicians to start precision immunotherapeutics that can prevent the development of severe, life-threatening complications and lasting disability in children.

## Supporting information

**S1 Table. TCR repertoire metrics of samples.**
(XLSX)

**S1 Fig. The effect of age, days after COVID-19 diagnosis and antibody levels on TRBV11-2 levels.** (a) The fraction of TRBV11-2 chains in patient repertoires as a function of the patients' age with Spearman's Rank Correlation coefficient (ρ) and P value. (b) The fraction of TRBV11-2 chains in patient repertoires as a function of the number of days after COVID-19 diagnosis with Spearman's Rank Correlation coefficient (ρ) and P value. (c) The fraction of TRBV11-2 chains in the repertoires of antibody negative and positive patients. Significance determined by unpaired Wilcoxon test between each paediatric group, with adjustment for multiple comparisons using Benjamini-Hochberg correction, indicated by: * p<0.05, ** p<0.01, and *** p<0.001. Lack of notation for specified comparisons indicates no statistical significance.
(TIF)

**S2 Fig. The effect of age, days after COVID-19 diagnosis and co-morbidities on the number of TCR clones in a repertoire.** (a) The number of clones/million reads in a repertoire as a function of the patients' age with Spearman's Rank Correlation coefficient (ρ) and P value. (b) The number of clones/million reads in a repertoire as a function of the number of days after COVID-19 diagnosis with Spearman's Rank Correlation coefficient (ρ) and P value. (c) The number of clones/million reads in the repertoires of patients with or without co-morbidities. Significance determined by unpaired Wilcoxon test between each paediatric group, with adjustment for multiple comparisons using Benjamini-Hochberg correction, indicated by: * p<0.05, ** p<0.01, and *** p<0.001. Lack of notation for specified comparisons indicates no statistical significance.
(TIF)

**S3 Fig. The fraction of MIRA clones in TCR repertoires.** (a) The fraction of class I MIRA clones in the repertoires of patient cohorts. Significance determined by unpaired Wilcoxon test between each paediatric group, with adjustment for multiple comparisons using Benjamini-Hochberg correction, indicated by: * p<0.05, ** p<0.01, and *** p<0.001. Lack of notation for specified comparisons indicates no statistical significance. (b) The number of class I MIRA clones as the function of the total number of clones in the patient repertoires with Spearman's Rank Correlation coefficient (ρ) and P value. (c) The fraction of class II MIRA clones in the repertoires of patient cohorts. Significance determined by unpaired Wilcoxon test between each paediatric group, with adjustment for multiple comparisons using Benjamini-Hochberg correction, indicated by: * p<0.05, ** p<0.01, and *** p<0.001. Lack of notation for specified comparisons indicates no statistical significance. (d) The number of class II MIRA clones as the function of the total number of clones in the patient repertoires with Spearman's Rank Correlation coefficient (ρ) and P value.
(TIF)

**S4 Fig. Distance to class I MIRA distribution of clones in the repertoires of children with mild COVID-19 or MIS-C.** Distance to class I MIRA distribution of clones in the repertoires of children with mild or MIS-C. Distance to class I MIRA distributions were fitted with Gaussian probability density functions and the difference between patient cohorts was assessed by comparing a model fitting all symptom groups together to a model taking into account the differences between symptom groups. The difference between the two models was determined with ANOVA analysis, with all p-values corrected for multiple hypothesis testing using Benjamini-Hochberg adjustment. Solid curves show the probability distribution functions determined by fitting the patient cohorts separately, shaded areas show 75% confidence intervals. P-values denote the statistical assessment of the models fitting all patient cohorts together versus fitting each patient cohort separately. See the distance to class I MIRA distribution of

TRBV11-2 clones in Fig 3b.
(TIF)

**S5 Fig. Distance to class II MIRA distribution of clones in the repertoires of children with mild COVID-19 or MIS-C.** Distance to class II MIRA distribution of clones in the repertoires of children with mild or MIS-C. Distance to class II MIRA distributions were fitted with Gaussian probability density functions and the difference between patient cohorts was assessed by comparing a model fitting all symptom groups together to a model taking into account the differences between symptom groups. The difference between the two models was determined with ANOVA analysis, with all p-values corrected for multiple hypothesis testing using Benjamini-Hochberg adjustment. Solid curves show the probability distribution functions determined by fitting the patient cohorts separately, shaded areas show 75% confidence intervals. P-values denote the statistical assessment of the models fitting all patient cohorts together versus fitting each patient cohort separately. See the distance to class II MIRA distribution of TRBV11-2 clones in Fig 3c.
(TIF)

## Author Contributions

**Conceptualization:** Amna Malik, Eszter N. Tóth, Michelle S. Teng, Margarita Dominguez-Villar, Stuart Adams, Nichola Cooper.

**Data curation:** Amna Malik, Eszter N. Tóth.

**Formal analysis:** Eszter N. Tóth.

**Funding acquisition:** Nichola Cooper.

**Investigation:** Amna Malik, Eszter N. Tóth.

**Methodology:** Eszter N. Tóth.

**Resources:** Eleanor Watt, Lauren Wise, Natalie Kent, Jack Bartram, Louis Grandjean, Nichola Cooper.

**Software:** Eszter N. Tóth.

**Supervision:** Michelle S. Teng, Jacob Hurst, Margarita Dominguez-Villar, Stuart Adams, Nichola Cooper.

**Visualization:** Eszter N. Tóth.

**Writing – original draft:** Amna Malik, Eszter N. Tóth.

**Writing – review & editing:** Amna Malik, Eszter N. Tóth, Michelle S. Teng, Margarita Dominguez-Villar, Nichola Cooper.

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
