## [Decision Letter · Decision Letter 0]

16 Jun 2022

PONE-D-22-08475Distorted TCR repertoires define multisystem inflammatory syndrome in childrenPLOS ONE

Dear Dr. Cooper,

Thank you for submitting your manuscript to PLOS ONE. After careful consideration, we feel that it has merit but does not fully meet PLOS ONE’s publication criteria as it currently stands. Therefore, we invite you to submit a revised version of the manuscript that addresses the points raised during the review process.

 1) After receiving the comments of one reviewer one big issue needs to be addressed in order to make this manuscript suitable for consideration in PLoS ONE.  While I assume that the omission of previous reports on the topic of this manuscript from the reference list was a mere oversight, including the respective references in the manuscript and discussing new data (obtained by this study) in the context of what was previously reported (missing references) is central to fulfil the criteria for publication.  Note that novelty is not a requirement and that sound and solid replication studies are also valued contributions.   2) All of the other points raised by reviwer #1 should also be appropriately and accurately addressed in a substatially revised resubmission.   

We look forward to receiving your revised manuscript.

Kind regards,

Sebastian D. Fugmann, Ph.D.

Academic Editor

PLOS ONE

Journal Requirements:

a) Did participants provide their written or verbal informed consent to participate in this study?

“N.C and M.D.V. are partially funded by Imperial College NIHR BRC. A.M. is funded by the Jon Moulton Charity.”

“E.N.T., M.S.T. and J.H. have a financial interest in Etcembly Ltd. The other authors declare no competing interest.”

7. We note that you have stated that you will provide repository information for your data at acceptance. Should your manuscript be accepted for publication, we will hold it until you provide the relevant accession numbers or DOIs necessary to access your data. If you wish to make changes to your Data Availability statement, please describe these changes in your cover letter and we will update your Data Availability statement to reflect the information you provide.

8. Please include captions for your Supporting Information files at the end of your manuscript, and update any in-text citations to match accordingly. Please see our Supporting Information guidelines for more information: http://journals.plos.org/plosone/s/supporting-information

Reviewers' comments:

Reviewer's Responses to Questions

**Comments to the Author**

1. Is the manuscript technically sound, and do the data support the conclusions?

Reviewer #1: No

2. Has the statistical analysis been performed appropriately and rigorously? 

Reviewer #1: N/A

3. Have the authors made all data underlying the findings in their manuscript fully available?

Reviewer #1: No

4. Is the manuscript presented in an intelligible fashion and written in standard English?

Reviewer #1: Yes

5. Review Comments to the Author

Reviewer #1: This is a new manuscript on the TCR repertoire analysis in patients with MIS-C.

A total of 20 children with COVID-19 have been included.

In the manuscript the authors state that “whether the specific T cell clones contribute to the hyperinflammatory state or if there is a difference in the T cell repertoire composition, antigen specificity is unknown”. This is not correct, and several studies have covered the repertoire of MIS-C patients on largest cohorts (Moreews et al. Science Immunol 2021, Hoste et al. JEM 2022, Sacco et al Nat Med 2022). The omission to refer to these papers in the manuscript and this statement are very surprising.

-The group of severity are poorly described and the “mild disease” comprise asymptomatic patients (that can thus not be defined as mild) and the severe group only comprise MIS-C patients (no other subset of ICU and pediatric COVID-19). This should be corrected, and the authors should clarify that they compare MIS-C to SARS-CoV2-positive children. The severe group is actually restricted to MIS-C. I suggest to avoid to name this group Severe COVID-19 (MIS-C instead). There are two severe manifestations of COVID-19 in children:

-Severe / fatal pneumonia, occurring in immunocompromised children (IFN-I pathway, in particular) and MIS-C. Here the authors only highlight the MIS-C phenotype.

-The use of DNA instead of RNA for the TCR sequencing is more challenging and associate a -greater number of biases.

-The junctional diversity of TRBV11-2 was also previously reported. In addition, it is present in both CD4/CD8 T cells and some functional studies have also tested T cell activation to various SARS-CoV2 antigens in MIS-C patients (Moreews et al. Hoste et al…).

-Actually the polyclonal expansion of T cells is a feature of superantigen immune reaction that was highlight in previous (omitted)studies. The other patients that experience an acute infection present an antigen-specific immunity that is also reported and physiological.

-The authors state that the TCR bias is responsible for the hyperinflammatory syndrome but this causal link is not supported by any data provided here.

-The timing to infection should be indicated. The so-called group “mild “ was possibly sampled at the acute phasis of the infection whereas the MIS-C are sampled at the time of the episode which is post-infectious (about 4 weeks after the virus encounter). This information is important.

-The clinical data on the MIS-C are missing (vasoplegia, blood pressure, shock, erythema…).

MIS-C usually occurs in patients with no comorbidities. How MIS-C was confirmed in the patient with hepatoblastoma ?

The TRBV11-2/ Vb21.3 expansion is now considered as a diagnosis marker. How was individual TRBV11-2 expansion (especially in the patient with hepatoblastoma)?

Is there any pediatric controls in the repertoire assessment?

In total, this is a replication study of numerous previous work on MIS-C (not severe pediatric COVID) and use a more challenging technique (DNA seq) to address this question on a lowest number of patients. The add-value of this work is limited compared to previous studies and the authors may provide information on how this report replicate previous work and what is new.

6. PLOS authors have the option to publish the peer review history of their article (what does this mean?). If published, this will include your full peer review and any attached files.

Reviewer #1: No

---

## [Author Response · Author response to Decision Letter 0]

4 Aug 2022

Comments from Reviewer #1: 

1. In the manuscript the authors state that “whether the specific T cell clones contribute to the hyperinflammatory state or if there is a difference in the T cell repertoire composition, antigen specificity is unknown”. This is not correct, and several studies have covered the repertoire of MIS-C patients on largest cohorts (Moreews et al. Science Immunol 2021, Hoste et al. JEM 2022, Sacco et al Nat Med 2022). Response: All these papers and other relevant papers have now been cited. 

2. The group of severity are poorly described and the “mild disease” comprise asymptomatic patients (that can thus not be defined as mild) and the severe group only comprise MIS-C patients (no other subset of ICU and pediatric COVID-19). This should be corrected, and the authors should clarify that they compare MIS-C to SARS-CoV2-positive children. The severe group is actually restricted to MIS-C. I suggest to avoid to name this group Severe COVID-19 (MIS-C instead). There are two severe manifestations of COVID-19 in children:

-Severe / fatal pneumonia, occurring in immunocompromised children (IFN-I pathway, in particular) and MIS-C. Here the authors only highlight the MIS-C phenotype.

Response: We have now changed the text in the manuscript to represent this. 

3. The use of DNA instead of RNA for the TCR sequencing is more challenging and associate a -greater number of biases.

Response: Although using DNA is more challenging and can lead to greater number of biases, the protocol we employed addresses these issues. The protocol used in this study has been standardised and used routinely in clinical setting for minimal residual disease (MRD) in cancer (Bartram et al.). Furthermore, analysis of the raw TCR sequences was performed using MiXCR which removes PCR errors before assembling clonotypes. Using DNA for bulk sequencing is also more quantitative, because there is one copy/cell, so the result will look more similar to e.g. single-cell sequencing, as opposed to RNA sequencing where you have varying number of RNA copies/cell. 

4. The junctional diversity of TRBV11-2 was also previously reported. In addition, it is present in both CD4/CD8 T cells and some functional studies have also tested T cell activation to various SARS-CoV2 antigens in MIS-C patients (Moreews et al. Hoste et al…).

Response: We have now added these references to the manuscript. 

5. Actually the polyclonal expansion of T cells is a feature of superantigen immune reaction that was highlight in previous (omitted)studies. The other patients that experience an acute infection present an antigen-specific immunity that is also reported and physiological.

Response: We have added the relevant references in the manuscript. 

6. The authors state that the TCR bias is responsible for the hyperinflammatory syndrome but this causal link is not supported by any data provided here.

Response: We agree with the reviewer and have changed the wording in the manuscript to represent that this is a speculation.

The timing to infection should be indicated. The so-called group “mild “ was possibly sampled at the acute phasis of the infection whereas the MIS-C are sampled at the time of the episode which is post-infectious (about 4 weeks after the virus encounter). This information is important.

Response: There was no significant difference in sampling between the two groups. The mild group was sampled at a median of 8 days post COVID-19 diagnosis and the MIS-C group was sampled at a median of 10 days post COVID-19 diagnosis. 

7. The clinical data on the MIS-C are missing (vasoplegia, blood pressure, shock, erythema…).

MIS-C usually occurs in patients with no comorbidities. How MIS-C was confirmed in the patient with hepatoblastoma ?

Response: Patient with hepatoblastoma was diagnosed based on having central nervous system disease, a clinical manifestation of MIS-C. We don’t have access to any further clinical data on the patients. 

8. The TRBV11-2/ Vb21.3 expansion is now considered as a diagnosis marker. How was individual TRBV11-2 expansion (especially in the patient with hepatoblastoma)?

Response: Patient with hepatoblastoma was diagnosed based on having central nervous system disease, a clinical manifestation of MIS-C. We don’t have access to any further clinical data on the patients. The spread of TRBV-12 expansion is shown on the box plot. The level of TRBV11-2 is significantly higher in MIS-C compared to controls; therefore, it is possible to say that it can be used as a biomarker. The spread in the MIS-C patients was quite even. 

9. Is there any pediatric controls in the repertoire assessment?

Response: There are paediatric control included in the repertoire analysis and referred to as ‘pre-COVID-19’ or ‘unexposed’ in the figures. 

10. In total, this is a replication study of numerous previous work on MIS-C (not severe pediatric COVID) and use a more challenging technique (DNA seq) to address this question on a lowest number of patients. The add-value of this work is limited compared to previous studies and the authors may provide information on how this report replicate previous work and what is new.

Response: Although the expansion and the polyclonality of TRBV11-2 have been shown before in MIS-C, we however, report the use of two metrics to define MIS-C in children infected with SARS-CoV-2: 1) CDR3-independent expansion of TRBV11-2+ T cells, 2) a lack of SARS-CoV-2 specificity in TRBV11-2+ T cells, measured by distance to Sars-Cov2-specific MIRA clones. These two metrics can serve as biomarkers for early detection of MIS-C guiding physicians to start precision immunotherapeutics that can prevent the development of severe, life-threatening complications and lasting disability in children. More specifically, the use of network embedding and the distance to MIRA analysis is the added value of our study.

---

## [Editor Report · Decision Letter 1]

25 Aug 2022

Distorted TCR repertoires define multisystem inflammatory syndrome in children

PONE-D-22-08475R1

Dear Dr. Cooper,

We’re pleased to inform you that your manuscript has been judged scientifically suitable for publication and will be formally accepted for publication once it meets all outstanding technical requirements.

Kind regards,

Sebastian D. Fugmann, Ph.D.

Academic Editor

PLOS ONE
---

## [Editor Report · Acceptance letter]

30 Sep 2022

PONE-D-22-08475R1 

Distorted TCR repertoires define multisystem inflammatory syndrome in children 

Dear Dr. Cooper:

I'm pleased to inform you that your manuscript has been deemed suitable for publication in PLOS ONE. Congratulations! Your manuscript is now with our production department. 

Kind regards, 

on behalf of

Dr. Sebastian D. Fugmann 

Academic Editor

PLOS ONE